# Theoretical Study at the Molecular Mechanics Level of the Interaction of Tetracycline and Chloramphenicol with the Antibiotic Receptors Present in *Enterococcus faecalis* (Q839F7) and *Streptococcus mutans* (Q8DS20)

**DOI:** 10.3390/antibiotics11111640

**Published:** 2022-11-16

**Authors:** Rufo Alberto Figueroa-Banda, Kimberly Francis Figueroa-Castellanos, Edith Angelica Chávez-Oblitas, María Elena Guillen-Nuñez, Flor Ayqui-Cueva, Bruno A. Del-Carpio-M, Karen L. Bellido-Vallejo, Badhin Gómez

**Affiliations:** 1Departamento de Odontología, Universidad Católica de Santa María, Urb. San José s/n—Umacollo, Arequipa 04013, Peru; 2Departamento de Ciencias Farmacéuticas, Bioquímicas y Biotecnológicas, Universidad Católica de Santa María, Urb. San José s/n, Umacollo, Arequipa 04013, Peru; 3Centro de Investigación en Ingeniería Molecular—CIIM, Universidad Católica de Santa María, Urb. San José s/n—Umacollo, Arequipa 04013, Peru

**Keywords:** antibiotic, *Streptococcus mutans*, *Enterococcus faecalis*, periodontitis, molecular dynamic simulation, MMPBSA, APBS

## Abstract

When dental infections occur, various types of antibiotics are used to combat them. The most common antibiotics to be used are tetracycline and chloramphenicol; likewise, the most common bacteria in dental infections are *Enterococcus faecalis* and *Streptococcus mutans*. In the present work, we have studied by molecular mechanics methods the interactions of the ribosomal proteins L16 present in *Enterococcus faecalis* and *Streptococcus mutans*, identified with UNIPROT code Q839F7 and Q8DS20, respectively. We evaluated the interactions between Q839F7 and Q8DS20 with tetracycline and chloramphenicol antibiotics. We found that the interaction between *Enterococcus faecalis* (Q839F7) is much more favorable when treated with chloramphenicol. In contrast, the interaction with tetracycline is favored in the case of Q8DS20 present in *Streptococcus mutans*. This suggests that the treatment should be differentiated depending on the infection level and the presence of some of these bacteria.

## 1. Introduction

Antibiotics, also known as drugs, are chemical compounds that interfere with the growth process of microorganisms, commonly used to treat bacterial infections [1,2]. In the dental field, antibiotics are considered beneficial for controlling intraoral bacteria, and an essential step for treating the oral cavity [3,4]. For example, the use of antibiotics is highly recommended in cases of severe caries that reach periapical inflammation [5], also in endodontic treatment, the elimination of bacteria by antibiotics is a fundamental step [6,7].

The symbiotic bacterial communities that they wish to eliminate are of the Gram-positive type, mostly [8], since these can adhere to the dentin of the root canal, creating an encapsulation with a self-produced matrix of extracellular polymeric compounds that results in the formation of a biofilm, becoming more resistant to antibacterial compounds [9,10]. Biofilms are the leading cause of primary and secondary root canal infections [11]. Within a mature biofilm, Actinomyces naeslundii, Lactobacillus salivarius, Streptococcus mutans [12,13,14,15,16], and *Enterococcus faecalis* [17,18,19,20] are commonly found [21,22].

*Streptococcus mutans* is the most prevalent microorganism in dental biofilm [23,24,25,26], this Gram-positive facultative anaerobic bacterium has the capacity for acidogenicity, aciduricity, acid tolerance, and ability to synthesize the extracellular matrix of the dental plaque [27,28,29]. Primary responsible for the generation of dental caries [30,31,32,33], the second most prevalent chronic disease worldwide [34] considered as the higher global health problem since it affects multiple groups old [35,36]. Another important microorganism is *Enterococcus faecalis*, a Gram-positive facultative anaerobic bacterium typical of the human intestinal microbiota but can be found infrequently in the oral cavity [30,37]. Although it is not a predominant bacterium, it has been identified as a microorganism frequently found in persistent or secondary endodontic infections in up to 77% [38,39] since it can penetrate dentin tubes and form biofilms along the root canal walls [40,41]. *Enterococcus faecalis* demonstrates the importance in cases of apical periodontitis due to its high resistance to unfavorable environments and intracanal medications that make it persist despite antimicrobial treatments [42,43].

The bacterial ribosome is a ribonucleoprotein particle with an approximate mass of 2.5 MDa whose function is to encode mRNA, made up of more than 50 proteins and three species of RNA organized into two subunits known as extensive or 50s and small or 30s, together they are called 70S [44,45]. The 30S subunit comprises 16S and 20 small proteins, while the 50S subunit comprises 23S and 5S and 35 small proteins [46]. The vital role of this organelle makes it an optimal target for antibiotic compounds since the failure of ribosomal function stops bacterial growth. An essential part of the ribosome is the ribosomal protein L16 of the large subunit of the ribosome. Harms et al. [47] determined a high-resolution structure of the large subunit of the Deinococcus radiodurans ribosome (D50S), the L16 located between two 23S rRNA stems, it has two crucial helices (38 and 89), and helix 38 generates a bridge between subunits in the 70S structure, helix 89 protrudes from the peptidyl transferase loop of 23S rRNA. Its root is in the center of peptidyl transferase. These rRNA stems and L16 form part of an aa-tRNA [48] binding site. The position of L16 Teraoka and Nierhaus [49] showed that L16 induces a significant conformational change in the subunit, suggesting that it organizes the architecture of a functional site in the subunit. In addition to its conformational role, Bashan’s group has recently found that L16 participates in ribosomal interactions with antibiotics because it has an exposed surface in the internal cavity of the ribosome [50]. The large subunit without the presence of the L16 protein is known to be defective in various activities, including interaction with antibiotics such as chloramphenicol, virginiamycin S, and erythromycin [51]. Experiments with mutations in the L16 protein in E. coli showed that L16 is directly involved in the activity of the translation machinery; the modification of the protein slows down bacterial growth [52]. Since this is an essential part of the bacterial growth mechanism, it makes L16 an exciting object of study.

Different antibiotics are used in dental treatments to eradicate the bacteria in the oral cavity. Among them, we have tetracycline, used in treatments against periodontitis as a microbial controller [53,54]. It is a broad-spectrum antibiotic with the ability to inhibit protein synthesis and is also administered orally with the ability to inhibit bacterial protein synthesis by binding to the 16S rRNA of the 30S ribosomal subunit [55,56]. Likewise, Chloramphenicol is also a potent broad-spectrum inhibitor, which has a high specific capacity for protein biosynthesis; its activity is based on reversible binding to the peptidyltransferase center in the ribosomal subunit 50S [57,58], among its effects is preventing the formation of peptide bonds, the termination of translation or reducing the precision of the translation [59]. This drug is not recommended due to its toxicity; in cases where the infection is resistant to conventional treatments, its use may be suggested [60].

As we have seen, the importance of protein L16 in bacterial protein synthesis is fundamental. Therefore, in the present study, we address the understanding of the interaction of L16 present in *Streptococcus mutans* and *Enterococcus faecalis* with a first antibiotic widely used in the treatment against bacteria, which is tetracycline, and a second antibiotic used in cases of resistance, such as chloramphenicol, analyzing the nature of molecular interactions, physicochemical and thermodynamic parameters, and energy calculations.

## 2. Computational Details

We searched for the ribosomal protein L16 present in *Streptococcus mutans* and *Enterococcus faecalis* in the UniProt database [61]. We found the corresponding proteins with codes Q8DS20 and Q839F7, respectively. We could access their PDB formats through the AlphaFlod server [62,63]. Likewise, we searched for the structures of the antibiotics tetracycline and chloramphenicol in the PubChem server; these we found with the CID codes: 54675776 and 5959. With the structures of both the ribosomal proteins and the antibiotics stabilized. Alternatively, we used the Gromacs computational package [64,65] to achieve its equilibrium structure in the case of proteins. We started by putting the proteins in a cubic box; we used the TIP3P solvent model [66] until the density was equal to one. Then, we neutralized them and added the effect of the saline ion at a concentration of 0.15 M; in both cases, we considered a physiological pH. Because the solvent effect is crucial, we decided to use the OPLS-AA force field [67] in our simulations; we applied periodic boundary conditions with a cut-off distance of 1.3 nm. Next, we proceeded to minimize the forces in double precision to avoid some atoms being very close; after the minimization, we proceeded to introduce the temperature to the system through the V-rescale thermostat [68], with a reference temperature of 309.65 ${}^{{}^{\circ}}$K (36.5 ${}^{{}^{\circ}}$C), we conducted through the NVT ensemble, for a time of 10 ns, after which the pressure we introduced employing the Berendsen barostat with a value of 1 Bar. Afterward, we ran in the NPT ensemble for a time of 500 ns. For antibiotics, we performed structure optimization using Gaussian 16 C.01 program [69], using a B3LYP hybrid functional with long-range correction using the Adaptive Coulombic Method (CAM) [70] and the TZVP basis set [71,72,73], with the final structures making use of the LigParGen server [74,75]. We generated the topology in the context of the OPLS-AA force field. With the structures of both the proteins and the antibiotics stabilized, we proceeded to dock them using the PathDock [76] and FireDock [77,78] servers. With the PathDock, the ten most probable structures were submitted to the FireDock server requesting 1000 docking events, as a refinement process of our structures. Obtaining our four most probable coupling systems, we proceeded to a molecular dynamics simulation for a time of 200 ns for each system in an NPT ensemble at the same temperature and pressure defined for the preliminary physiological systems. We performed various analyses based on the trajectory generated by the molecular dynamics simulation; we calculated the binding energy using the MMPBSA computational package [79,80]. Likewise, we performed the electrostatic potential analysis using the APBS server [81] and the Ramachandran diagrams from the PDBSum server [82]. We used the UCSF Chimera [83], GaussView [84], and Gnuplot [85] programs for visualization purposes.

## 3. Results and Discussion

By searching the UniProt server, we found the proteins linked to the interaction regions with antibiotics corresponding to *Streptococcus mutans* (Q8DS20) and Enterococcus feacalis (Q839F7) at the ribosomal level. The complete structures for both cases we generated utilizing the AlphaFold server. Afterward, we stabilized the systems using a molecular dynamics simulation in two stages: the first to introduce the effect of temperature in a canonical ensemble (NVT) for a simulation period of 10ns; after that, we searched for the structure in the balance, under conditions of temperature, pressure and saline ion effect, since these parameters are the ones that we consider most important in natural conditions of the organism, for which we simulated molecular dynamics in an Isobaric-Isothermal ensemble, for a 500 ns time.

For the case of Enterococcus feacalis (Q839F7), in Figure 1a, we present the root mean square deviation analysis generated at each step of the molecular dynamics simulation, also known as RMSD. By analyzing the graph, we can see that the structure of our protein entered the equilibrium phase in the last 300 ns; when the delta is 0.2, we consider that the system is in the equilibrium zone. In Figure 1b, we present the fluctuation of the residues during the entire trajectory of the molecular dynamics simulation; we can notice that between residues 71 to 100 approximately it offers movement, and at the ends of the structure, which is reasonable since the extremes always present a greater degree of freedom in their direction. In Figure 1c, we show the radius of gyration, which is typically associated with the compaction process of the system, in many cases, is referential for its stabilization.

However, we can see that hydrogen bonds, in the first 125 ns, tend to decrease in value; this subsequently increases between 150 ns and 350 ns and then increases again from 450 to 500 ns, which indicates that the protein, despite being in an equilibrium zone, tends to compact, decreasing its volume. Whenever the hydrogen bonds increase, we can associate it with a decrease in the volume occupied by the protein. In Figure 1, we show the evolution of the hydrogen bonds along the trajectory. We can see that these practices remain in equilibrium, which indicates that the protein structure is balanced because if it presented a loss drastic of the hydrogen bridges, it would be denatured, a case that does not occur in our simulation of molecular dynamics.

Additionally, we show in Figure 2a the analysis of the electrostatic potential for Q839F7; we present the protein at the beginning and end of the molecular dynamics simulation. Likewise, we showed two images of the front and back of the protein; we can notice that already from initiation, the structure gives a full positive surface charge, as can be seen in the blue areas, but after the simulation process, this is accentuated fundamentally in the upper part of the protein system. On the other hand, in Figure 2b, the surface is presented in a transparent form of the electrostatic potential to appreciate the positions of the secondary structure better; from this, we can quickly realize that the terminal end presents movement concerning the initial configuration. Additionally, from Figure 2c, we can observe the appearance of beta sheets, fundamentally between amino acids 12–14 and between 32–36, the sections that present a secondary structure of alpha helix, which remains constant during the simulation process, in summary, we can say that our protein system has gained secondary structure slightly.

We can then say from the graphs of the RMSD, RMSF, Rg, and Hydrogen Bonds that our system is stabilized, only presenting a movement in its terminal region, so this does not affect the general stability of the system, fundamentally due to these regions not usually being associated with active sites.

We stabilized the Q8DS20 protein belonging to the *Streptococcus mutans* under the same temperature, pressure, and saline ion effect conditions as our other system. Thus, in Figure 3, we present the RMSD analysis noting that the Q8DS20 reaches equilibrium quickly; this occurs at approximately 200 ns. Furthermore, when we analyze the graph corresponding to the fluctuation suffered by the amino acid residues during the simulation of molecular dynamics (see Figure 3b), we can observe that it presents a minor change in the N-terminal region and maintains the fluctuation on the C-terminal region, and as in the previous case, amino acids from 70 to 95 present a significant change. In Figure 3c, we show the radius of gyration, and we see that from 250 ns, a compression process of the protein begins. In the case of hydrogen bonds, we can observe that there is a fluctuation between gain and loss of the same, but which, in general terms, remains constant throughout the trajectory (see Figure 3d).

In Figure 4, we present the electrostatic potential analysis (see Figure 4a), where we can observe that its nature is highly positive superficially, which is accentuated after the simulation of molecular dynamics to achieve the equilibrium phase. Protein systems usually contain positively and negatively charged residues; some are neutral or polar. When the APBS approach calculates a classical electrostatic potential, amino acid residues are often ordered, generating regions or zones with specific electrostatic characteristics. For the case of the secondary structure (see Figure 4b), we do not observe a more significant type of variation in the chain and; therefore, in the secondary form, which is consistent with its rapid arrival at the equilibrium zone. On the other hand, in the structure analysis results during the molecular dynamics simulation trajectory, we observed that beta sheets were lost between amino acids 71 and 75 of the protein (see Figure 4c).

In determining the isoform present at equilibrium, the protein structure of Q839F7 is resistant to reaching equilibrium because it exhibits a more significant fluctuation in the RMSD compared to protein Q8DS20. This fluctuation is not decisive as to the nature of its activity. Still, it may indicate that it is probably an effect of the medium and may be associated with its diffusion capacity in the medium.

With the structures of the Q839F7 and Q8DS20 proteins, we proceeded to generate the interacting systems using an assembly; we conducted this with the PathDcok server; the docking performed was blind since we wanted to explore the entire surface of the proteins to find the areas of the most significant probability of interaction with antibiotics, the server run performed a search for the best 1000 complexes. After that, we ran a refinement of the 1000 structures through the FireDock server, which conducted an exploration of a thousand events for each system without considering a seed speed; therefore, it maintains the blind search system and reports a total of 20 structures, which have the best interaction energy for each of the cases.

Likewise, it is essential to mention that, in general, the first five structures of the couplings are found in the same cavity but with slight conformational changes of the ligand, which are considered conformational changes of antibiotics. When we resolve calculations of the molecular docking, usually, the force fields used are not the exact ones used in molecular dynamics simulations; it is for this reason that, to be consistent, the values reported by the servers are only referential but not determinative. If we consider these values, it can give us a wrong idea of which of the antibiotics is the one that best interacts with the protein. In addition, when we observe the attractive van der Waals term, we can see that tetracycline is more favorable than chloramphenicol. Still, in the repulsive van der Waals terms, we keep that chloramphenicol is the one that presents a better value compared to tetracycline. Still, it is pretty remarkable that tetracycline is approximately twice the value of chloramphenicol at the contact atomic energy. Likewise, we have to consider that the coupling process is nothing more than a probabilistic event, and the energies must be deterministic; therefore, it is necessary to perform a molecular dynamics simulation of the system; we have accomplished this simulation for a time equal to 200 ns.

In Table 1, we present the twenty results that offer the best interaction energy; the energy is reported in kcal/mol but is only referential since it depends on the force field used by the FireDock server. The results presented correspond to both antibiotics (tetracycline and chloramphenicol) interacting with the protein Q839F7. In the case of the interaction with tetracycline, it is the complex structure number 474, the one that presents the highest global energy of interaction, with a value of −40.01 kcal/mol, while in the case of chloramphenicol, it is complex structure number 3, with a global interaction energy of −29.09 kcal/mol.

In Figure 5a, we can observe that the RMSD, the complex of Q839F7 with chloramphenicol, reaches an equilibrium condition much quicker during its interaction process. In contrast, in the case of tetracycline, we can observe a significant fluctuation, fundamentally between 60 and 170 ns, after which it seems to hold steady. We have reviewed the molecular dynamics simulation files between 80 and 160 ns, and we have observed that the antibiotic rearranges itself, rotating on the surface of the protein, which generates the variation of the RMSD, but at no time is it seen to move away from the system. When we analyze the change by residues during the molecular dynamics simulation (see Figure 5b), we notice that the interaction stabilizes in both cases. Because the residues that previously fluctuated when Q839F7 was alone have already lost their mobility, we can only see that in the case of interaction with tetracycline, the C-terminal region shows much more significant fluctuation than in the case of chloramphenicol. In Figure 5c, we present the radius of gyration; in the case of chloramphenicol, this remains more or less constant.

In contrast, in the case of tetracycline, there is a wide fluctuation during the simulation; we can say that, in general, it tends to decompress to protein. In the case of the analysis of hydrogen bonds (see Figure 5d) for chloramphenicol, we can observe that they remain constant, fluctuating between the loss and gain of hydrogen bonds. Still, in general terms, it remains stable, unlike tetracycline; we can see that between 60 ns, there is a significant loss of hydrogen bonds, the same as at 160 ns.

From this, we deduced that in the case of Q839F7, there was more stabilizer interaction with chloramphenicol than tetracycline. Still, we must conduct more rigorous energy analyses for a conclusive study. In Figure 6, we can see that the site of interaction of chloramphenicol with the Q839F7 protein occurs fundamentally in amino acids of one of the alpha helix chains, and it is lateral; the interacting amino acids are Met1, Leu2, Arg6, Asp44, Arg45, Pro70, Gly93, and Trp93, we can notice that its composition has a robust favorable charge composition. It is probably due to the interaction between the positively charged cavity and the accumulation of negative charge on the atoms of the chloramphenicol molecule, thus generating an exchange governed by electrostatic forces.

For the case of the interaction between Q839F7 with tetracycline, we showed it in Figure 7. Where we can see that the binding site has changed drastically, now found in the upper part of the protein but in the entirely positive area, as well as in the previous case, when we observe the interaction from the top and expand to identify the amino acid residues that generate the exchange. These are Phe13, Gly88 and Ala89, which is only three interactions. In comparison, in the previous case, there were eight interactions; additionally, we did not find positively charged amino acids in these residues. However, the cavity is of a positive electrostatic nature. The fact that tetracycline presents an unstable interaction is probably due to the interaction site and the reduced number of interactions with the amino acid residues of Q839F7.

As indicated above, for a conclusive observation that the antibiotic is more favorable to interact with Q839F7, it requires more sophisticated calculations, and we perform the analysis of the interaction energy throughout the molecular dynamics simulation trajectory using the MMPBSA approximation. We presented the results in Table 2. In this approach, the energy components were divided for the various non-bonding interactions linked to calculating the interaction energy.

We can see that in the case of electrostatic energy, in the case of tetracycline, the exchange is positive, which we can interpret as non-contributory. In contrast, that same energy component is harmful in the case of chloramphenicol. With a high value, the interaction energy is generally much more favorable for chloramphenicol than tetracycline. We can now conclusively indicate that in the case of Q839F7, it prefers to interact with chloramphenicol than with tetracycline.

In the case of Q8DS20 present in *Streptococcus mutans*, we carried out the docking with the two antibiotics (tetracycline and chloramphenicol) on the PathDock server. We used the Firedock server to refine the results of PathDock. Table 3 shows the results. Of the refining process of the docking models, so we can note that for the case of interaction with tetracycline, model number 245 is the best energetically (−42.02 kcal/mol), while for the case of chloramphenicol, model number 1 is the best according to global energy (−31.04 kcal/mol).

Additionally, we can note that the highest negative value for the attractive van der Waals interactions is much higher for the case of tetracycline compared to that of Chloramphenicol. Still, the terms associated with the van der Waals repulsion are lower in the case of Chloramphenicol. When considering the atomic contact energy (ACE) of Chloramphenicol compared to tetracycline, we look at the ACE as more favorable for tetracycline than Chloramphenicol. Preliminarily, we could indicate without being conclusive that, in the case of Q8DS20, tetracycline will be a better candidate for a favorable interaction. Still, we require more exhaustive analyses for a definitive assertion.

For the complex models 245 with tetracycline and 1 for chloramphenicol, we proceeded to perform a molecular dynamics simulation for a time of 200 ns. Below, we discuss the properties linked to stabilization. Figure 8a shows the RMSD for the interactions of tetracycline and chloramphenicol with Q8DS20. We can observe that tetracycline generally is more stable during the entire trajectory of the molecular dynamics simulation, while chloramphenicol presents some abrupt movements between 92 and 116 ns and again between 140 and 160 ns, after which it stabilizes. Figure 8b shows the amino acid residues’ fluctuation during the molecular dynamics simulation trajectory.

In both cases, the models note that they contribute to structural stabilization or that they anchor the structure so that it loses mobility. When we analyze the radius of gyration (see Figure 8c), we observe that tetracycline contributes to interactions leading to the packing of Q8DS20. At the same time, chloramphenicol generates a volume gain between 100 to 120 ns and later between 145 and 160 ns, then returning to constant values. In Figure 8d, we can see the hydrogen bonds; in both cases, it remains constant, so we could not say there is a gain or loss of hydrogen bonds during the molecular dynamics simulation time.

In Figure 9, we present the structure of the interaction complex between chloramphenicol and Q8DS20; the preferred region seems to be the base of the protein, and this area does not show an accumulation of surface charges. When we enlarge it, we observe that it only interacts with the amino acid residues Arg60, Gly61, Asp108, and Pro110; of these residues, the only one that contributes with a positive charge is Arg60. The Asp108 residue neutralizes this; we could infer the weak interaction.

We visually present the interactions between tetracycline and Q8DS20 in Figure 10; we can see that the interaction position has changed towards an electropositive part. When analyzing the amino acid residues that interact with tetracycline, we find that there are five: Arg45, Gly88, Ala89, Pro90, and Glu91, of which we can see that Arg45 confers a positive capacity to the region. However, if a Glu91 is present, it is separated from the positive part. We infer that because both charged amino acids are far apart in the protein amino acid chain, we find that the interaction with tetracycline is favorable in this case.

In Table 4, we present the interaction energy for tetracycline and chloramphenicol with the Q8DS20, obtained from the MMPBSA approximation for the entire trajectory of the molecular dynamics simulation. In this case, the interaction with tetracycline is almost double that of chloramphenicol. The most significant contributor to this result is the energy due to van der Waals interactions, and in this parameter, we can see that tetracycline is much higher than chloramphenicol.

## 4. Conclusions

We have found stable structures for the proteins: Q839F7, present in *Enterococcus faecalis*, and Q8DS20, present in *Streptococcus mutans*; both proteins correspond to the ribosomal zone of interaction with antibiotics. The antibiotics used in the present study were tetracycline and chloramphenicol. From this, we found that in the case of protein Q839F7, the interactions favor chloramphenicol since its interaction site contains eight amino acid residues. However, in the case of protein Q8DS20, the interactions tend to be tetracycline, presenting a total of five amino acid residues with which it interacts. The interaction preference is supported by the calculation of the interaction energy using the MMPBSA approximation, which in both cases, is consistent with the results of the interactions. We can infer that treatment with chloramphenicol is much more favorable in the presence of *Enterococcus faecalis*. In contrast, for the existence of *Streptococcus mutans*, treatment with tetracycline will be much more efficient. 

## Figures and Tables

**Figure 1 antibiotics-11-01640-f001:**
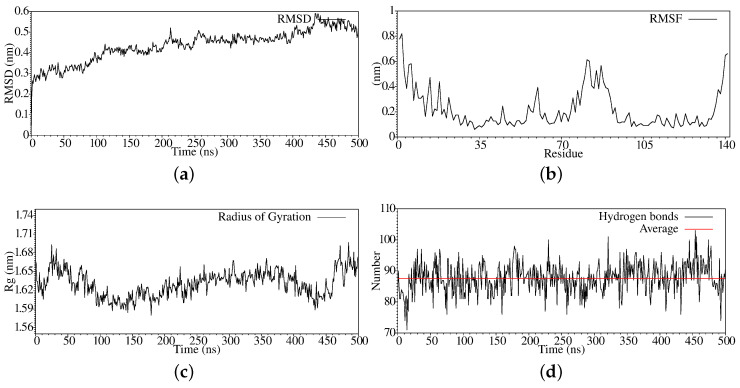
Stabilization properties of Q839F7 after 500 ns molecular dynamics simulation: (**a**) RMSD; (**b**) RMSF; (**c**) Radius of Gyrate; (**d**) Hydrogen bonds.

**Figure 2 antibiotics-11-01640-f002:**
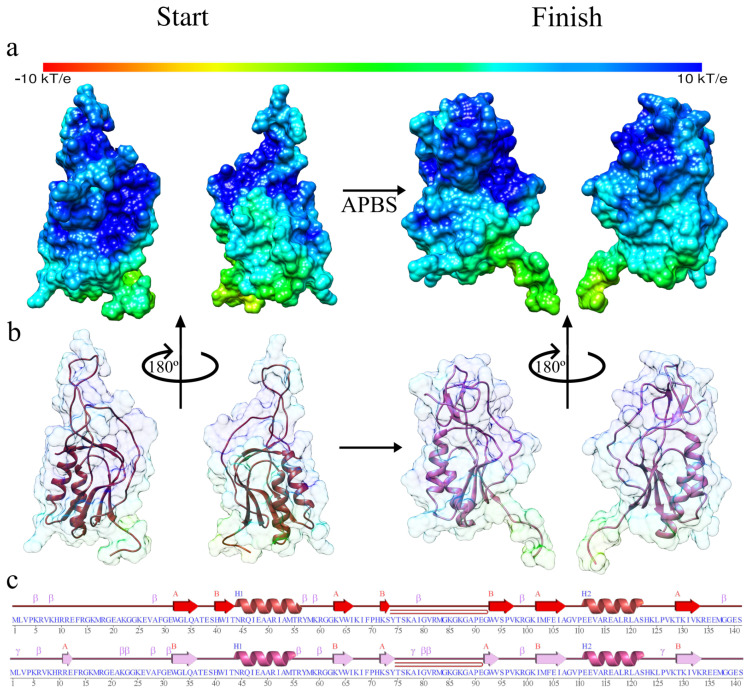
The structure of the Q839F7 in the molecular dynamics simulation at the beginning and end: (**a**) Electrostatic potential at the beginning and end of the molecular dynamics simulation; (**b**) Attenuated surface of the APBS to show the secondary structure at the beginning and end of the molecular dynamics simulation; (**c**) Analysis of the secondary structure at the beginning and end of the molecular dynamics simulation derived from the PDBSum.

**Figure 3 antibiotics-11-01640-f003:**
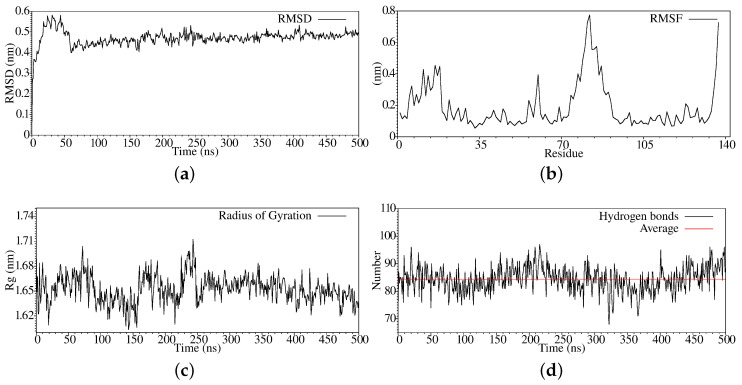
Stabilization properties of Q8DS20 after 500ns molecular dynamics simulation: (**a**) RMSD; (**b**) RMSF; (**c**) Radius of Gyrate; (**d**) Hydrogen bonds.

**Figure 4 antibiotics-11-01640-f004:**
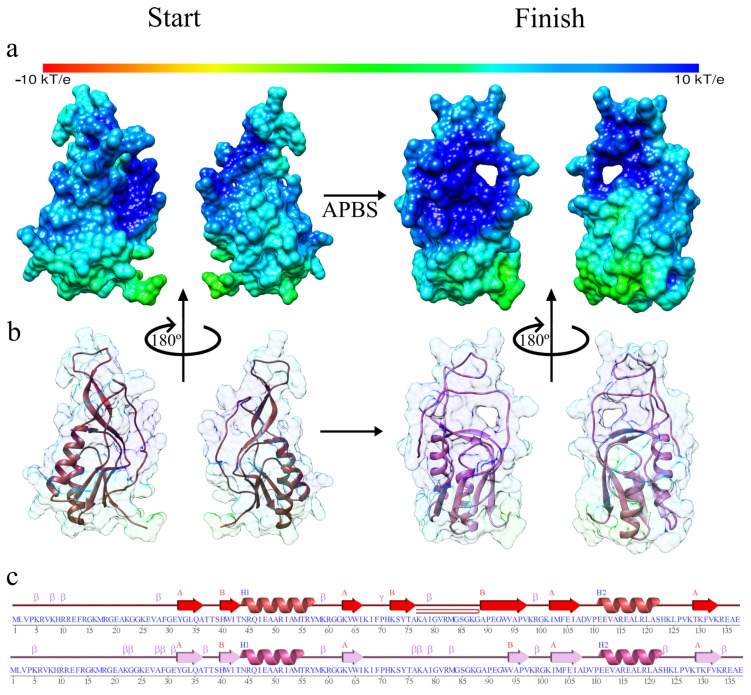
The structure of the Q8DS20 in the molecular dynamics simulation at the beginning and end: (**a**) Electrostatic potential at the beginning and end of the molecular dynamics simulation; (**b**) Attenuated surface of the APBS to show the secondary structure at the beginning and end of the molecular dynamics simulation; (**c**) Analysis of the secondary structure at the beginning and end of the molecular dynamics simulation derived from the PDBSum.

**Figure 5 antibiotics-11-01640-f005:**
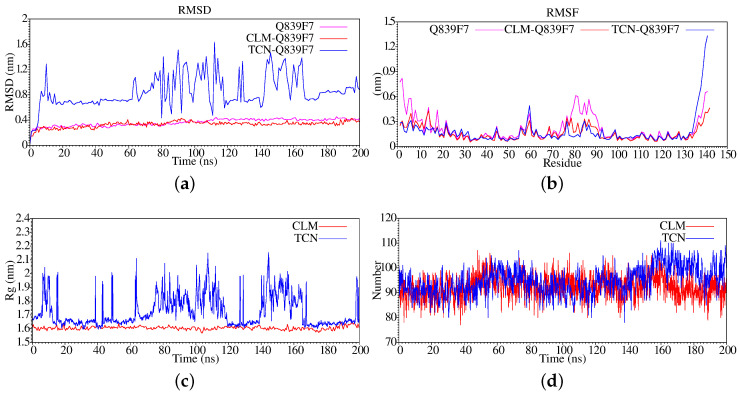
Stabilization properties of Q839F7 with the Inhibitors after 200 ns molecular dynamics simulation: (**a**) RMSD; (**b**) RMSF; (**c**) Radius of Gyrate; (**d**) Hydrogen bonds.

**Figure 6 antibiotics-11-01640-f006:**
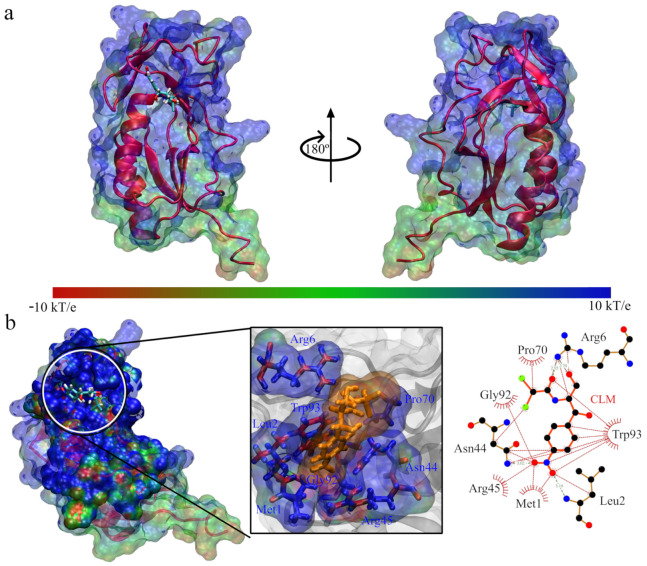
Interaction between Q839F7 and chloramphenicol: (**a**) Front and side view of the interacting system Q839F7 with chloramphenicol; (**b**) View of the interaction site and enlargement, accompanied by the interactions between the site and chloramphenicol.

**Figure 7 antibiotics-11-01640-f007:**
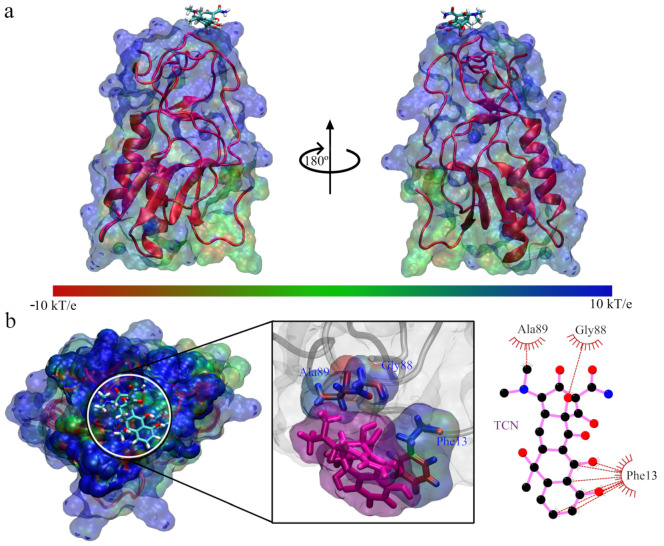
Interaction between Q839F7 and tetracycline: (**a**) Front and side view of the interacting system Q839F7 with tetracycline; (**b**) View of the interaction site and enlargement, accompanied by the interactions between the site and tetracycline.

**Figure 8 antibiotics-11-01640-f008:**
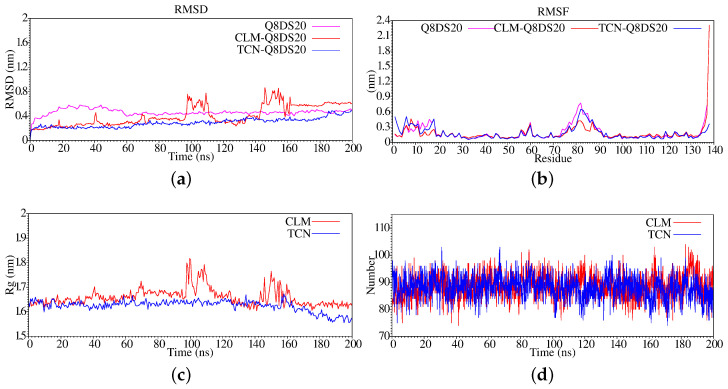
Stabilization properties of Q8DS20 with the Inhibitors after 200 ns molecular dynamics simulation: (**a**) RMSD; (**b**) RMSF; (**c**) Radius of Gyrate; (**d**) Hydrogen bonds.

**Figure 9 antibiotics-11-01640-f009:**
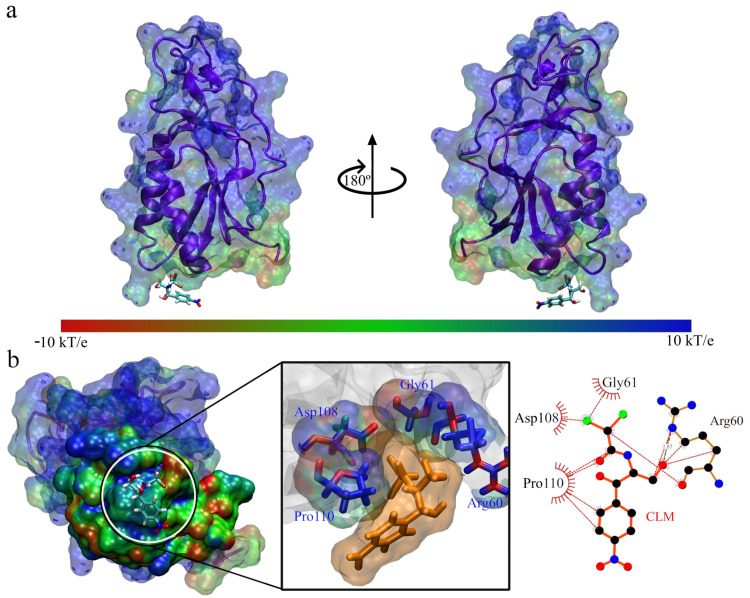
Interaction between Q8DS20 and chloramphenicol: (**a**) Front and side view of the interacting system Q839F7 with chloramphenicol; (**b**) View of the interaction site and enlargement, accompanied by the interactions between the site and chloramphenicol.

**Figure 10 antibiotics-11-01640-f010:**
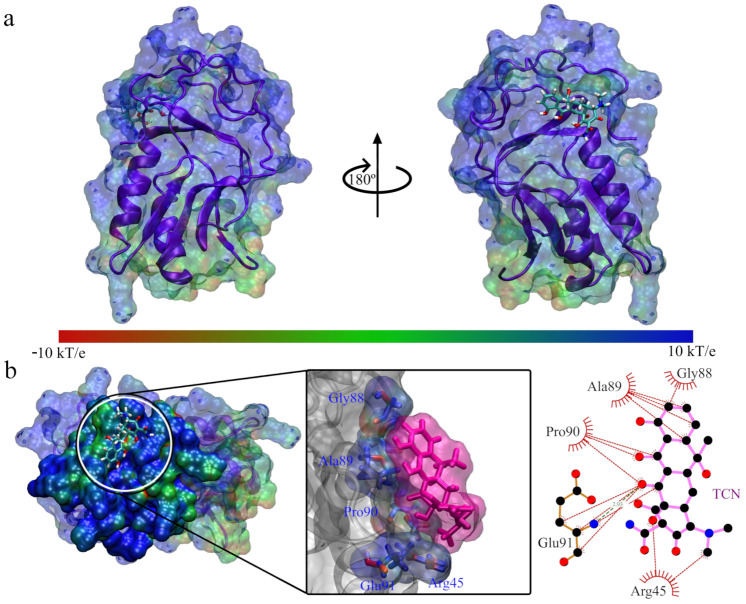
Interaction between Q8DS20 and tetracycline: (**a**) Front and side view of the interacting system Q839F7 with tetracycline; (**b**) View of the interaction site and enlargement, accompanied by the interactions between the site and tetracycline.

**Table 1 antibiotics-11-01640-t001:** We present the twenty most important results from the Firedock server for coupling Q839F7 with tetracycline and chloramphenicol.

Num ${}^{a}$	Tetracycline				Num ${}^{a}$	Chloramphenicol			
	GE ${}^{b}$	A–VdW ${}^{c}$	R–VdW ${}^{d}$	ACE *^e^*		GE ${}^{b}$	A–VdW ${}^{c}$	R–VdW *^d^*	ACE *^e^*
474	−40.01	−17.74	3.36	−13.29	3	−29.09	−14.26	2.24	−7.06
1	−34.79	−16.65	3.46	−9.80	80	−28.62	−14.36	5.97	−8.54
223	−34.04	−19.00	5.59	−8.31	76	−27.54	−13.89	5.54	−9.65
291	−32.02	−16.66	2.89	−9.20	358	−27.18	−14.02	2.34	−7.40
52	−31.80	−19.18	3.10	−8.23	40	−26.60	−14.11	6.24	−7.41
122	−31.77	−18.38	5.31	−7.87	12	−26.30	−12.71	2.23	−7.37
105	−31.76	−15.96	7.41	−10.92	70	−26.16	−13.70	2.34	−6.71
9	−31.21	−20.30	9.59	−7.56	248	−26.01	−14.60	4.60	−6.26
207	−31.12	−16.06	4.91	−11.11	61	−25.20	−13.41	1.62	−6.58
242	−30.79	−15.63	5.62	−10.82	114	−24.47	−13.95	6.36	−8.84
61	−30.46	−17.88	6.50	−8.27	2	−24.36	−12.50	5.43	−7.47
47	−30.07	−15.12	1.59	−7.21	69	−24.19	−15.98	10.85	−7.55
10	−29.41	−17.88	7.38	−8.35	128	−24.02	−12.92	2.04	−8.14
8	−29.28	−17.51	3.64	−7.33	35	−24.01	−13.96	6.98	−7.73
49	−29.27	−19.22	5.16	−7.79	181	−23.57	−12.30	3.36	−7.74
25	−29.26	−16.60	1.66	−6.19	34	−23.43	−12.17	3.93	−7.55
21	−29.02	−16.59	5.81	−7.74	18	−23.37	−13.30	2.61	−5.42
140	−28.85	−13.43	2.50	−10.29	6	−23.33	−13.93	5.06	−6.20
196	−28.73	−15.02	5.76	−9.95	354	−23.20	−14.84	10.85	−7.25
155	−28.31	−16.03	4.91	−8.35	78	−23.03	−11.30	4.89	−8.78

*^a^* Solution number of Firedock; *^b^* Global Energy in kcal/mol; *^c^* Attractive VdW in kcal/mol; *^d^* Repulsive VdW in kcal/mol. *^e^* Atomic Contact Energy in kcal/mol. Appendix A, Figure A1 and Figure A2 present the structures of the top five highest-ranked complexes according to global energy for tetracycline and chloramphenicol.

**Table 2 antibiotics-11-01640-t002:** The binding energy of the interaction of the bacterium *Enterococcus faecalis* (Q839F7) with both antibiotics.

Energy Component (kJ/mol)	Q839F7— *Enterococcus faecalis*	
	Tetracycline	Chloramphenicol
van der Waals energy	−68.250 ± 27.109	−84.947 ± 15.319
Electrostatic energy	25.815 ± 51.636	−77.417 ± 29.606
Polar solvation energy	17.606 ± 23.842	66.502 ± 20.757
SASA energy	−9.084 ± 3.356	−10.697 ± 1.295
SAV energy	−74.343 ± 51.103	−87.315 ± 16.921
Binding energy	−108.256 ± 80.005	−193.874 ± 33.456

**Table 3 antibiotics-11-01640-t003:** We present the twenty most important results from the Firedock server for coupling Q8DS20 with tetracycline and chloramphenicol.

Num ${}^{a}$	Tetracycline				Num ${}^{a}$	Chloramphenicol			
	GE ${}^{b}$	A–VdW ${}^{c}$	R–VdW ${}^{d}$	ACE *^e^*		GE ${}^{b}$	A–VdW ${}^{c}$	R–VdW ${}^{d}$	ACE *^e^*
245	−42.02	−23.66	10.83	−10.43	1	−31.04	−16.42	1.56	−5.45
167	−41.94	−20.82	7.00	−12.44	263	−30.13	−15.74	1.90	−5.67
541	−39.70	−19.62	6.90	−11.04	259	−29.31	−15.00	5.39	−8.54
19	−39.32	−19.30	1.98	−8.27	4	−29.31	−17.13	4.92	−4.76
14	−38.64	−19.81	4.18	−8.39	5	−28.65	−13.49	1.78	−6.44
144	−37.24	−19.19	2.95	−9.41	190	−28.55	−14.66	5.51	−10.88
5	−37.06	−18.26	2.28	−8.65	8	−28.33	−13.38	4.48	−7.91
50	−36.98	−18.15	3.13	−10.84	27	−27.65	−13.80	3.35	−9.21
107	−36.71	−19.65	4.59	−10.34	6	−27.42	−13.48	1.11	−6.55
208	−36.71	−19.70	2.66	−9.27	73	−27.32	−14.64	1.02	−5.28
9	−36.71	−19.95	2.72	−6.77	20	−27.22	−14.90	5.07	−7.52
39	−36.55	−17.88	2.58	−8.92	101	−27.14	−14.94	5.67	−8.59
12	−36.38	−19.83	3.48	−7.48	33	−27.05	−14.75	4.16	−5.88
20	−36.32	−19.38	7.90	−9.76	24	−26.47	−13.89	6.36	−7.53
2	−36.30	−23.43	7.92	−5.14	51	−26.10	−15.79	6.83	−7.52
181	−35.58	−18.06	10.45	−14.43	12	−25.97	−13.78	5.96	−6.78
26	−35.34	−19.66	5.45	−7.10	135	−25.88	−13.83	6.72	−9.00
349	−34.79	−20.13	11.52	−11.71	47	−25.73	−11.88	2.89	−8.63
8	−34.75	−21.05	4.38	−5.11	13	−25.61	−16.05	4.38	−5.06
302	−34.71	−19.42	11.17	−13.14	171	−25.58	−15.33	4.49	−6.71

*^a^* Solution number of Firedock; *^b^* Global Energy in kcal/mol; *^c^* Attractive VdW in kcal/mol; *^d^* Repulsive VdW in kcal/mol. *^e^* Atomic Contact Energy in kcal/mol. Appendix A, Figure A3 and Figure A4 present the structures of the top five highest-ranked complexes according to global energy for tetracycline and chloramphenicol.

**Table 4 antibiotics-11-01640-t004:** The binding energy of the interaction of the bacterium *Streptococcus mutans* (Q8DS20) with both antibiotics.

Energy Component (kJ/mol)	Q8DS20—*Streptococcus mutans*	
	Tetracycline	Chloramphenicol
van der Waals energy	−148.927 ± 22.604	−61.400 ± 35.589
Electrostatic energy	2.548 ± 42.704	−24.125 ± 35.802
Polar solvation energy	61.456 ± 19.352	34.231 ± 31.575
SASA energy	−16.754 ± 1.733	−8.074 ± 4.459
SAV energy	−146.648 ± 23.330	−62.666 ± 42.733
Binding energy	−248.326 ± 48.584	−122.035 ± 71.319

## Data Availability

Not applicable.

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
