# Peer review of "Theoretical Study at the Molecular Mechanics Level of the Interaction of Tetracycline and Chloramphenicol with the Antibiotic Receptors Present in Enterococcus faecalis (Q839F7) and Streptococcus mutans (Q8DS20)"

_antibiotics, 2022, doi:10.3390/antibiotics11111640_

Round 1
Reviewer 1 Report
1. The abstract, methodology and results sections need to be improved. Particularly the manner in which these parts are presented, calls for a scientific-style write-up.
2. The manuscript contains a number of spelling and grammatical mistakes.
3. Major improvements must be made in how global stability indexes (Line 142-158, 163-174 ) are interpreted.
4. Give an explanation to the following statements
a. Line 176-178
b. line 197-203
5. In figure 5,the fluctuation pattern that appears on the rmsf graph is not supported by the results of rmsd.
5. In order to comprehend the fluctuating pattern that develops once the antibiotics occupy the active site, it is also more logical to compare the bound and free proteins in a single graph (both rmsf and rmsd).
6. The authors must thoroughly visualise the trajectory file to see what happens to the protein structure bound to tcn between 80 and 160ns.
7. To support the mmpbsa findings, it's better to add the residue wise energy contributions to unravel the role of individual residues to the overall binding
8. Discuss why the simulation time period was 500ns for the free protein while for the bound state it is 200ns?
9. Table 3. headings need revision (spelling mistakes), also Van der walls should be replaced by van der Waals
Author Response
Author's Reply to the Review Report (Reviewer 1)
1.- The abstract, methodology and results sections need to be improved. Particularly the manner in which these parts are presented, calls for a scientific-style write-up.
We have revised the summary, methodology, and results section so that they present an excellent scientific style; we appreciate your recommendations.
- The manuscript contains a number of spelling and grammatical mistakes.
We have conducted an exhaustive review of the spelling and grammar of the manuscript, and we have proceeded to correct the existing errors; these were fixed in the text.
- Major improvements must be made in how global stability indexes (Line 142-158, 163-174 ) are interpreted.
The necessary corrections have been made, and we have included a small summary paragraph.
- Give an explanation to the following statements
- Line 176-178
We added in text for clarity.
- line 197-203
We add in the text to improve understanding.
- In figure 5, the fluctuation pattern that appears on the rmsf graph is not supported by the results of rmsd.
The RMSD measures the variation of the structure compared to the previous step, but it is a measure of the protein as a whole. Usually, it is plotted against the time of the trajectory of the molecular simulation. The RMSF gives us information on the fluctuation of each of the residues during the trajectory of the molecular simulation, and it is usually plotted against the number of amino acid residues. Therefore, one gives us information on how much the protein moves in general, and the second on which regions are the ones that fluctuate the most during the molecular dynamics simulation.
- In order to comprehend the fluctuating pattern that develops once the antibiotics occupy the active site, it is also more logical to compare the bound and free proteins in a single graph (both rmsf and rmsd).
We believe their observation is adequate for better clarity of the discussion, so we have added them to the corresponding figures. However, we have not discussed them because they are already in the part where we analyze the stabilized structures alone.
- The authors must thoroughly visualise the trajectory file to see what happens to the protein structure bound to tcn between 80 and 160ns.
We observed that it occurs due to a movement of the antibiotic concerning the surface of the protein. Still, for practical purposes, the antibiotic does not move away from the interaction zone, so we did not see it as necessary to emphasize that observation. Nevertheless, due to the statement, we added a few lines about it.
- To support the mmpbsa findings, it's better to add the residue wise energy contributions to unravel the role of individual residues to the overall binding
We believe that analyzing the energy contribution by residues would be fascinating. Still, the purpose of our research is more general, to know if both antibiotics act similarly or if there is some differentiation in their administration process in a specific dental infection, so for future work. Still, we do not believe it is necessary for the conclusions found in our research.
- Discuss why the simulation time period was 500ns for the free protein while for the bound state, it is 200ns?
In the literature, much more time is usually used to achieve a structure in equilibrium than an interacting structure in the zone equilibrium, and this value is generally around 300 to 500ns. We have proteins obtained by prediction methods using artificial intelligence; we believe that it requires a longer molecular dynamics simulation time to achieve the equilibrium phase; for this reason, we used 500ns in the molecular dynamic simulation; once getting this starting structure to perform the molecular dockings, a shorter time is needed to accomplish the interacting equilibrium zone, therefore that 200ns is the most prudent, they usually use 100ns, but we believe that with 200ns we are observing the interaction phenomenon properly.
- Table 3. headings need revision (spelling mistakes), also Van der walls should be replaced by van der Waals
We revised the header of table 3 and realized the changes recommended.
Reviewer 2 Report
In the work of Rufo Alberto Figueroa-Banda et al. The author applied docking and MD simulations to study the preference of interactions between the ribosome proteins from pathogenic bacteria with antibiotics. They identified Enterococcus faecalis ribosome protein (code Q839F7) is more favorable to interact with chloramphenicol, and the ribosome protein Q8DS20 presented in mutant Streptococcus is favored to interact with tetracycline, suggesting the treatment for dental infections should be differentiated.
For docking results of Tetracycline and Chloramphenicol with the ribosome protein code Q839F7, the overall binding energy of Tetracycline with Q839F7 is more favorable than the complexes of Chloramphenicol with Q839F7. The authors used MD simulation to validate these docking results. However, they compare only the complex with the highest binding energy in both groups (complex 474 with complex 3). How about the other complexes (1,223,291, 52, 122,105, 9, 207 with complex 3)? Are they also less stable than complex 3 during the MD simulation? The authors should show these data to support their conclusion.
Author Response
Author's Reply to the Review Report (Reviewer 2)
In the work of Rufo Alberto Figueroa-Banda et al. The author applied docking and MD simulations to study the preference of interactions between the ribosome proteins from pathogenic bacteria with antibiotics. They identified Enterococcus faecalis ribosome protein (code Q839F7) is more favorable to interact with chloramphenicol, and the ribosome protein Q8DS20 presented in mutant Streptococcus is favored to interact with tetracycline, suggesting the treatment for dental infections should be differentiated.
For docking results of Tetracycline and Chloramphenicol with the ribosome protein code Q839F7, the overall binding energy of Tetracycline with Q839F7 is more favorable than the complexes of Chloramphenicol with Q839F7. The authors used MD simulation to validate these docking results. However, they compare only the complex with the highest binding energy in both groups (complex 474 with complex 3).
How about the other complexes (1,223,291, 52, 122,105, 9, 207 with complex 3)?
When molecular docking methods are used to obtain preliminary interacting structures, it is essential to consider two critical factors; all docking prediction methods use stochastic processes since they are repeated for many events, usually 1000, then from it, the servers report the most probable structures. The researcher must decide what type of structure to use since the expected interaction region may not be the one selected by the server. Still, in the case of not knowing the interaction region, the best structure is only blindly considered to be able to continue with the study since if we believe the other systems, it will generally take a very long time to solve the molecular dynamics simulations, for this reason, we have only considered the most probable form indicated by the server. However, the energies reported by the server are indeed very close; this is not usually what is obtained after a simulation of molecular dynamics of the interacting system; for this reason and practicality, it is only recommended to use the most probable; we emphasize this since we do not know a specific region of interaction, we cannot prefer any particular area. Likewise, the first five structures were very close to each other; they were only conformational changes of the antibiotics. This strengthens our selection of a single system, the one with the best score in molecular coupling.
Are they also less stable than complex 3 during the MD simulation?
A priori reviewing the energies reported by the server where the molecular coupling was performed as a trend, which we should be able to find during the molecular dynamics simulation, typically, this trend is maintained; the only thing that varies is the value of the interaction energy, under this assumption derived from the empirical observations of other researchers, is that we do not perform more simulations than the best structure provided by the server. Still, we cannot be conclusive as to whether they are less stable, but if we consider that the trends are maintained, it is most likely that the same thing will happen; that is, the trend will continue, but the energy values will be different.
The authors should show these data to support their conclusion.
For the reasons stated above, we do not believe they are necessary for this study.
Round 2
Reviewer 1 Report
Q1. The title of the manuscript needs rectification to reflect the content of the studies
Q2. The abstract part should be a brief elaboration of the background, and then put forward your own methods for the study, and finally a brief summary of the experimental part with findings.
Q3. My question remains the same: the fluctuation pattern on the rmsd resonates around 1.2nm for TCN compared to the other two results (averaging around 0.2nm), but when we look at the rmsf results retrieved from the same trajectory for which the rmsd analysis was done, the fluctuation pattern for all three systems is practically identical. Kindly explain?
Q4. Please cite more references from the last three years
Q5. Table 3. heading still needs revision (Tetraciclina, Cloranfenicol)
Author Response
Q1. The title of the manuscript needs rectification to reflect the content of the studies
The Text says:
Theoretical Study of the Interaction of Tetracycline and Chloramphenicol with the Receptors of Antibiotics in the Enterococcus faecalis and Streptococcus mutans
The Text will say:
Theoretical Study at the Molecular Mechanics level of the Interaction of Tetracycline and Chloramphenicol with the Antibiotic Receptors present in Enterococcus faecalis (Q839F7) and Streptococcus mutans (Q8DS20)
Q2. The abstract part should be a brief elaboration of the background, and then put forward your own methods for the study, and finally a brief summary of the experimental part with findings.
The Text says:
In the present work, we have studied by theoretical methods the interactions of proteins present in Enterococcus faecalis and Streptococcus mutans; identified according to the literature were Q839F7 and Q8DS20, present in the ribosome of bacteria, respectively. For this, we consider using two antibiotics: tetracycline and chloramphenicol. When dental infections occur, these antibiotics are used; in the case of chloramphenicol, it is used as a last option. We have found that the interaction between Enterococcus faecalis Q839F7 is much more favorable when treated with chloramphenicol. In contrast, in the case of Q8DS20 present in Streptococcus mutans, the interaction with tetracycline is favored; Depending on the level of infection and the presence of any of these bacteria, the treatment should be differentiated.
The Text will say:
When dental infections occur, various types of antibiotics are used to combat them. The most common antibiotics to be used are tetracycline and chloramphenicol; Likewise, the most common bacteria in dental infections are Enterococcus faecalis and Streptococcus mutans. In the present work, we have studied by molecular mechanics methods the interactions of the ribosomal proteins L16 present in Enterococcus faecalis and Streptococcus mutans, identified with UNIPROT code Q839F7 and Q8DS20, respectively. We evaluated the interactions between Q839F7 and Q8DS20 with tetracycline and chloramphenicol antibiotics. We found that the interaction between Enterococcus faecalis (Q839F7) is much more favorable when treated with chloramphenicol. In contrast, the interaction with tetracycline is favored in the case of Q8DS20 present in Streptococcus mutans. This suggests that the treatment should be differentiated depending on the infection level and the presence of some of these bacteria.
Q3. My question remains the same: the fluctuation pattern on the rmsd resonates around 1.2nm for TCN compared to the other two results (averaging around 0.2nm), but when we look at the rmsf results retrieved from the same trajectory for which the rmsd analysis was done, the fluctuation pattern for all three systems is practically identical. Kindly explain?
The RMSD and the RMSF are different measurements of molecule properties; in the case of the RMSD, it measures how the structure changes between one step and the next; the RMSD is always a single value for the entire system. Therefore, this own point of the step in which we are is continuously graphed concerning the time in which we are in the trajectory. In the case of the RMSF, it is an average measure of how a residue or an atom has been moving throughout the simulation. For this reason, we should not expect correspondence between the RMSF and the RMSD. We can see that the RMSD can be compared with all those properties that depend directly on the trajectory times, as is the case of the radius of gyrate.
Q4. Please cite more references from the last three years
We have added references as indicated by the referee.
Q5. Table 3. heading still needs revision (Tetraciclina, Cloranfenicol)
We fixed the translation error.
Reviewer 2 Report
"Likewise, the first five structures were very close to each other; they were only conformational changes of the antibiotics. " The author should make this statement in the manuscript, and show the data in the supplementary. How about the complex with no. 9 and 207? are they also only conformational changes of the antibiotics? The ensemble of the docking complexes should be shown in the supplementary.
Author Response
"Likewise, the first five structures were very close to each other; they were only conformational changes of the antibiotics. " The author should make this statement in the manuscript, and show the data in the supplementary.
The following text was added to the body of the manuscript.
“Likewise, it is essential to mention that, in general, the first five structures of the couplings are found in the same cavity but with slight conformational changes of the ligand, which are considered conformational changes of antibiotics.”
How about the complex with no. 9 and 207? are they also only conformational changes of the antibiotics?
The solutions marked as 9 and 207 for tetracycline with Q839F7 are not in the same position nor conformational isomers. Still, we must remember that in this type of approximation, in which stochastic methods are used, it is essential to consider that the energy criterion is mandatory; for this reason, we only use the structure with the highest probability and the lowest energy.
The ensemble of the docking complexes should be shown in the supplementary.
We have added the figures that show the first five solutions of the molecular coupling of each of the systems as an appendix.